# MicroRNA Alterations Induced in Human Skin by Diesel Fumes, Ozone, and UV Radiation

**DOI:** 10.3390/jpm12020176

**Published:** 2022-01-28

**Authors:** Giuseppe Valacchi, Erika Pambianchi, Simona Coco, Alessandra Pulliero, Alberto Izzotti

**Affiliations:** 1Animal Science Department, Plants for Human Health Institute, North Carolina State University, Research Campus Kannapolis, Kannapolis, NC 28081, USA; gvalacc@ncsu.edu (G.V.); epambia@ncsu.edu (E.P.); 2Department of Environmental Sciences and Prevention, University of Ferrara, 44121 Ferrara, Italy; 3Department of Food and Nutrition, Kyung Hee University, Seoul 130-701, Korea; 4Lung Cancer Unit, IRCCS Ospedale Policlinico San Martino, 16132 Genova, Italy; simona.coco@hsanmartino.it; 5Department of Health Sciences, University of Genova, 16132 Genova, Italy; alessandra.pulliero@unige.it; 6Department of Experimental Medicine, University of Genova, 16132 Genova, Italy; 7UOC Mutagenesis and Cancer Prevention, IRCCS San Martino Hospital, 16132 Genova, Italy

**Keywords:** microRNAs, environmental risk factors, cutaneous tissues, ozone exposure

## Abstract

Epigenetic alterations are a driving force of the carcinogenesis process. MicroRNAs play a role in silencing mutated oncogenes, thus defending the cell against the adverse consequences of genotoxic damages induced by environmental pollutants. These processes have been well investigated in lungs; however, although skin is directly exposed to a great variety of environmental pollutants, more research is needed to better understand the effect on cutaneous tissue. Therefore, we investigated microRNA alteration in human skin biopsies exposed to diesel fumes, ozone, and UV light for over 24 h of exposure. UV and ozone-induced microRNA alteration right after exposure, while the peak of their deregulations induced by diesel fumes was reached only at the end of the 24 h. Diesel fumes mainly altered microRNAs involved in the carcinogenesis process, ozone in apoptosis, and UV in DNA repair. Accordingly, each tested pollutant induced a specific pattern of microRNA alteration in skin related to the intrinsic mechanisms activated by the specific pollutant. These alterations, over a short time basis, reflect adaptive events aimed at defending the tissue against damages. Conversely, whenever environmental exposure lasts for a long time, the irreversible alteration of the microRNA machinery results in epigenetic damage contributing to the pathogenesis of inflammation, dysplasia, and cancer induced by environmental pollutants.

## 1. Introduction 

The World Health Organization (WHO) estimated that circa 90% of the global urban population lives with pollutant levels exceeding WHO guideline limits. This has been linked to the premature death of seven million people each year [1]. The target organs of pollution include the lungs, gut, brain, and mainly the skin [2].

The skin is the largest sensory organ (approximately 2 m^2^) in our body and is composed of two main layers: the epidermis and the dermis. The dermis is mainly formed of fibroblasts involved in the secretion of elastin and collagen fibers, embedded with nerve endings, sebaceous glands, hair follicles, and blood and lymphatic vessels. During the process of differentiation/keratinization, keratinocytes withdraw from the cell cycle and begin to express differentiation-dependent markers (i.e., keratins), eventually becoming anucleated densely keratinized corneocytes [3]. These cells are held together in the multilayered stratum corneum by a lipid-laden extracellular matrix (ECM), which performs the barrier function of the skin [4]. Recently, it has been demonstrated that the skin is not an impenetrable tissue, and it can even be a gateway for certain pollutants, even affecting internal organs [5].

The use of the word “pollution” can be misleading given that there are several different pollutants that can affect our health. Based on their chemical and physical properties as well as their sources, The United States Environmental Protection Agency (EPA) has identified the most common air pollutants, also known as “criteria air pollutants”, as ozone (O_3_), particulate matter (PM), carbon monoxide (CO), lead, sulfur dioxide (SO_2_), and nitrogen dioxide (NO_2_) [6]. Clear evidence of the correlation between each pollutant and skin disorder has not yet been established; however, the harmful effects of O_3_, PM, and UV radiation have been well demonstrated. To date, only a few studies have compared the cutaneous effect of different pollutant exposure. Several skin diseases such as atopic dermatitis (AD), psoriasis, acne, and, in some cases, also skin cancer have been linked, either directly or indirectly, to pollution exposure, although the debate is still open. In particular, exposure to O_3_ and PM have been demonstrated to be associated with skin aging, including wrinkle formation and dark spots, respectively [7]. In addition, the study by Xu et al. [8] demonstrated the association between ozone exposure and cutaneous conditions by analyzing the emergency room (ER) visits for skin conditions together with the levels of several air pollutants such as O_3_, PM_10_, SO_2_, and NO_2_. The authors were able to extrapolate that skin conditions such as urticaria, eczema, contact dermatitis, rash/other non-specific eruptions, and infected skin diseases were exacerbated by exposure to increased ozone levels. Another more recent publication has further examined the association of short-term changes in air quality with emergency department (ED) visits for urticaria in Canada. A total of 2905 ED visits were analyzed, and a positive and significant correlation was observed between air quality levels and ED visits for urticaria, confirming that air pollution can affect skin physiology [9,10].

O_3_ and PM have quite different mechanisms of action, while O_3_ is not able to penetrate the skin and reacts directly with the lipids present in the stratum corneum, PM can possibly enter the skin via the hair follicles or enable the skin to absorb components present in the PM (such as polycyclic aromatic hydrocarbons—PAH) and lead to an epidermal OxInflammatory reaction [11,12].

Indeed, it is now well established that ozone and diesel particles, together with UV radiation, can induce a proinflammatory response in parallel to an altered tissue redox homeostasis [13].

In addition to the ability of pollution to produce oxidative and inflammatory mediators, recent studies have indicated that DNA methylation patterns can be greatly influenced by environmental factors such as ambient air pollution, and these epigenetic changes are linked with diverse diseases [14,15,16].

In fact, several reports have shown that epigenetic alterations could be an important pathway through which environmental factors exert their effects [14,17]. Epigenetic refers to the alterations in gene expression levels that occur without changes in the underlying DNA sequence (such as DNA methylation, histone modification, miRNA, and noncoding RNA expression [18,19]). It should be mentioned that several pathologies, including cancers, have been associated with epigenetic modifications [18]. Exposure to environmental stimuli may result in epigenetic changes, which can impact gene expression and predisposition to developing pathological conditions. [20]. Understanding epigenetic alterations due to exposure to specific pollutants may lead to the development of biomarkers to assess the disease risk associated with air pollution. Micro-RNAs and noncoding RNAs (ncRNAs) play critical roles in gene expression and contribute to epigenetic control in the process [21]. For this reason, the present study aimed to evaluate the different miRNA epigenetic patterns related to the specific exposure of cutaneous tissues to pollutants such as ozone, diesel exhaust, and to the stressor UV radiation, which is the most toxic and most present in urban areas.

## 2. Materials and Methods

### 2.1. Ex Vivo Human Skin Explants Preparation

Human skin biopsies (12 mm diameter) were obtained from three healthy Caucasian donors (40–45 years old) who underwent elective abdominoplasties at Hunstad/Kortesis/Bharti Cosmetic Surgery clinic. In total, 24 punch biopsies were taken from the abdominal skin of each donor, and subcutaneous fat was removed with sterile scissors and a scalpel.

The biopsies, comprising dermal and epidermal layers, were rapidly rinsed with Phosphate-Buffered Saline (PBS, Gibco, New York, NY, USA). The biopsies were then moved into 6-well dishes containing 2 mL of complete Dulbecco’s Modified Eagle Medium (DMEM) with 10% Fetal Bovine Serum (FBS) and 1% of antibiotics and antimycotics (100 U/mL penicillin and 100 µg/mL, Gibco, New York, NY, USA) added; then, they were incubated at 37 °C in 5% CO_2_ for overnight recovery.

The following day the medium was replaced with a fresh one, and the biopsies were exposed to the different pollutants as discussed below.

The experiment was performed at least in triplicate for each condition and donor.

### 2.2. Ex Vivo Human Skin Explants Ozone (O_3_) Exposure 

A full 24 h after skin biopsy collection, ex vivo explants were allocated into a plexiglass sealed chamber, connected to the ozone generator machine (ECO3 model CUV-01, Model 306 Ozone Calibration Source, 2B Technologies, Ozone solution, ITA), and exposed to 0.2 ppm for 4 h. Sample biopsies were then collected following ozone exposure (T0) or after 24 h (T24).

### 2.3. Ex Vivo Human Skin Explants Diesel Engine Exhaust (Diesel) Exposure 

Another set of skin biopsies was exposed to diesel engine exhaust by letting the engine run for 10 s and allowing the exhaust to reach the sealed exposure chamber where the skin biopsies remained for 30 min. Specifically, the skin explants were placed into a sealed plexiglass box connected to a Kubota RTV-X900 diesel engine (3-cylinder, 4-cycle diesel with overhead valves, 1123 cc with 24.8 HP at 3000 rpm). After the 30 min of Diesel exposure, the exposure medium was changed with a fresh one, and the biopsies were either collected (T0) or moved back into the incubator at 37 °C in 5% CO_2_ for 24 h (T24).

### 2.4. Ex Vivo Human Skin Explants Ultraviolet Light (UV) Exposure 

The other human skin biopsies were exposed to 200 milli Joule (mJ) UVA/UVB light, which equates to circa 2 h at solar apex and corresponding to 10 minimal erythemal doses (MED, 1 MED = 20 mJ/cm^2^) [22]. UVA/UVB light (exposure of circa 20 s) was generated by a Sol1A Class ABB Solar Simulator, equipped with a xenon lamp (Newport Oriel Sol1A, CA, USA). Samples were collected after UV exposure (T0), or after 24 h (T24).

To match the real solar spectrum at the condition of the sun at the Zenith angle of 0, we performed UV exposure with a UVA/UVB ratio of 21:1 measured with a radiometer ILT2400 Hang-Held Light Meter/Optometer (International Light Technologies, Inc., Peabody, MA, USA).

### 2.5. Total RNA Extraction and Lyophilization

Total RNA extraction was performed using a miRNeasy Mini kit QIAGEN (Hilgen, DE, cat. 1038703) and Qiazol Lysis Reagent 50 QIAGEN (Hilgen, DE, cat. 1023537), according to the manufacturer’s protocol. Briefly, skin biopsies were homogenized in 700 µL of Qiazol Lysis Reagent, with a tissue homogenizer (Precellys 24 homogenizer, 5 cycles 6500 rpm 3 × 30 s, at 4 °C). Samples were then centrifuged (12,700 rpm, 5 min at 4 °C), the supernatant was collected and transferred to a new tube containing 140 µL of Chloroform, and then centrifuged again (12,000 g, 15 min at 4 °C). The upper aqueous phase was transferred to a new tube containing 1.5 volumes of ethanol 100% and mixed thoroughly. Then, half of the volume was moved into the RNeasy Mini spin column and centrifuged (8000× *g*, 30 s, RT). This last step was repeated for the other half of the volume. Next, 700 µL of diluter Buffer RWT was added to the column, centrifuged (8000× *g*, 30 s, RT), and then the liquid was discarded. Next, the addition of 500 µL of diluted Buffer RPE to the column, centrifugation (8000× *g*, 30 s, RT), and discarding of the liquid was repeated twice. Finally, the columns were moved into new Rnase free tubes, centrifuged at maximum speed for 1 min, and 30 µL of RNase free water was added to the spin column membrane. Elution of RNA was performed by centrifugation at 8000 g, 1 min, RT.

Then, the eluted RNA was concentrated via a lyophilization process (1.5 h, Low Setting, RT) using a Savant DNA SpeedVac Concentrator (Savant DNA120, DNA 120 OP, Thermo Electron Corporation, Waltham, MA, USA).

### 2.6. miRNA-Microarray and Bioinformatic Analyses

miRNA expression profiling was carried out by an Agilent platform, following the miRNA Microarray protocol v.3.1.1 (Agilent Technologies, Santa Clara, CA, USA). Briefly, 50 ng of total RNA containing miRNAs and Spike-in controls underwent dephosphorylation and a labeling step with Cyanine 3-pCp. The Cy3-labeled RNA was then purified using Micro Bio-Spin P-6 Gel Column (Bio-Rad Laboratories, Inc., Hercules, CA, USA) and hybridized on a Human miRNA microarray slide 8 × 60K (Agilent Technologies; including 2549 miRNAs, miRBase 21.0) at 55 °C for 20 h. After washing, the slides were scanned by a G2565CA scanner (Agilent Technologies), and the images were extracted by Feature Extraction software v.10 (Agilent Technologies). Microarray raw data were deposited in the Gene Expression Omnibus (http://www.ncbi.nlm.nih.gov/geo); GEO number accession requested, 15 March 2022).

Bioinformatic analyses were performed using the GeneSpring software (GeneSpring Multi-Omic Analysis v 14.9 by Agilent Technologies). For each sample, the intensities of replicated spots on each array were log transformed and averaged. Data processing was performed by 3D principal component analysis (PCA) scores and Hierarchical Clustering.

Comparisons between sets of data were performed by evaluating the fold changes. A volcano Plot T-Test analysis for all miRNA entities was run, using Fold Change ≥2 and *p*-value ≤ 0.05. Because log transformed data were used, negative and zero signals were transformed into 0.01 values. This approach could result in artificially high-fold variations. To correct this artifact, we now report in Tables that fold variation values upregulated more than 10 times into ‘>10-fold’ and fold variation values downregulated more than 10 times into <0.1-fold.

miRNAs related to three different environmental exposures (Environmental Exposure miRNA Signature) were determined by analyzing miRNAs comparatively in exposed vs. non-exposed subjects. Environmental exposures were determined for each sample according to (a) Diesel, (b) Ozone, and (c) UV.

To understand the relationship between environmental exposure signatures and their biological significance in human tissues, a target detection for each environmental exposure signature was performed using the TargetScan prediction database.

## 3. Results 

### Comparison of miRNA Expression Profile between Pollutants

The overall trend of miRNA expression in human skin either untreated or exposed to diesel, ozone, and UV was evaluated at 0 and 24 h by Line plot analysis. (Figure 1).

The expression line plot was similar at 0 and 24 h in unexposed skin. Diesel induced dramatic alteration of miRNA profile compared to air but mainly after 24 h. Conversely, miRNA alteration induced by ozone was remarkable at time 0 while being much more attenuated compared to untreated skin at 24 h. miRNA profiles were slightly increased by UV exposure, mostly at 24 h compared to unexposed skin.

Scatter plot analyses were performed to assess the number of miRNAs with more than two-fold profile alteration compared to untreated (air-exposed), Figure 2.

Each miRNA is represented by colored dots, whose expression intensity can be inferred from the position on the horizontal and vertical axes. The horizontal axis indicates the miRNA expression level in untreated samples and the vertical axis in treated (diesel, ozone, and UV) samples. The central diagonal lines indicate the equivalence (<two-fold variation) in the intensity of miRNA expression in treated as compared to untreated samples. miRNA dots falling outside the green diagonal lines indicate higher than two-fold differences in miRNA expression between the tested experimental conditions. Scatter plot analyses compared untreated (air) and treated (diesel, ozone, and UV) samples both at 0 and 24 h. miRNA colors reflect the signal intensity in the treated samples (red is high, yellow is intermediate, and blue is low). Upregulated miRNAs are located in the upper-right area, and downregulated miRNAs are in the lower-right area of the scatter plots.

For diesel and to a lesser extent for UV, a cloud of downregulated miRNAs was detected already at 0 h, while the additional cloud of upregulated miRNAs was detected only at 24 h.

Conversely, for ozone, two clouds of both upregulated and downregulated miRNAs were already detected at time 0.

At time 0, out of the 2549 miRNAs tested, 100 (2.5%) were downregulated (blue dots) while 122 (4.8%) were upregulated (red dots) after exposure to diesel. Immediately after ozone exposure, we evidenced 262 (10.3%) downregulated miRNAs and 297 (11.6%) upregulated; however, upon UV exposure, we detected that 238 (9.3%) were downregulated and 294 (11.5%) upregulated.

At 24 h, out of the 2549 miRNAs tested, 219 (8.6%) were downregulated (blue dots), and 251 (9.8%) were upregulated (red dots) after exposure to diesel; 229 (8.9%) were downregulated, and 238 (9.3%) upregulated after ozone exposure; and 174 (6.8%) were downregulated and 241 (9.4%) upregulated after UV exposure.

The effects of the tested pollutants on the whole miRNA expression profile were compared by unsupervised principal component analysis of variance (PCA) and supervised hierarchical cluster (HCA) analyses.

The PCA at 0 h (Figure 3) showed that the miRNA profiles of ozone and UV treated samples were remarkably altered, being located far away and in another quadrant, as compared to the untreated (air) samples. Conversely, the miRNA profile in diesel-treated samples was only slightly distant from the untreated samples, being located in the same quadrant.

The PCA at 24 h (Figure 3) reported that the miRNA profiles of all pollutant-treated samples (including diesel) were located far away and in another quadrant compared to the untreated sample. The samples treated with diesel and ozone were close to each other but far away from the UV-treated sample. This finding indicates that the pattern of miRNAs altered by UV is quite different from those induced by ozone and diesel exposure. This situation is likely due to the different pathogenic mechanisms induced by exciting radiation (UV) as compared to gaseous (ozone) and mixed gaseous-particulate pollutants (diesel).

The HCA at 0 h (Figure 4) showed that the most remarkable alterations in miRNA profiles were induced by ozone, whose expression profile was located at the right of the hierarchical tree far away from the untreated (air) sample. An intermediate situation occurred for UV, whose alteration profile was in the central part of the hierarchical tree. miRNA alterations induced by diesel were less remarkable; indeed, the profile was linked to the untreated (air) sample in the hierarchical tree.

The HCA at 24 h (Figure 4) indicated that the most remarkable alterations in miRNA profiles were induced by diesel, whose expression profile was located at the right of the hierarchical tree far away from the untreated (air) sample. An intermediate response was visible for UV, whose alteration profile was located in the central part of the hierarchical tree. On the other hand, the changes in miRNA profiles induced by ozone exposure were less remarkable; in fact, this profile was linked to the untreated (air) sample in the hierarchical tree.

A Venn diagram data representation was used to identify microRNAs presented in both lists, i.e., altered by each pollutant both at 0 and 24 h.

These miRNAs represent the specific miRNA signature induced by each pollutant. Their identity is reported in Table 1 (diesel), Table 2 (ozone), and Table 3 (UV). These Tables enlist miRNAs modulating their expression more than two-fold and above the statistical significance threshold of *p* < 0.05 considering the four replicates spotted in each microarray. A comparison of fold variation was made by dividing the signal intensity detected in treated skin by those detected in untreated skin. Fold variation values >2.0 indicate upregulation after treatment and <0.5 downregulation after treatment. Available information for the main biological pathways regulated by modulated microRNAs is also reported (column Function), as well as the reference from where this information was collected.

MicroRNAs reported in Table 1, Table 2 and Table 3 were selected by both volcano plot analysis and Venn diagram data representation. These miRNAs represent the specific signature of each pollutant, being modulated both after short-term exposure (UV 20 s, diesel 30 min, and ozone 4 h) and long-term exposure (24 h for all pollutants).

Fold variations reported in Table 1, Table 2 and Table 3 are the rate of signal intensity changes between treated and untreated samples at 24 h.

The main biological pathways regulated by these miRNAs, as inferred from available literature, are also reported.

The main pathways targeted by tested pollutants were: apoptosis, cell cycle, and inflammation (Table 4).

## 4. Discussion and Conclusions 

The constant exposure to oxidants, including ultraviolet (UV) radiation and other environmental pollutants, such as diesel fuel exhaust and ozone, makes the skin our first defense against the outdoor environment, and it is also the tissue more affected by outdoor stressors. The contribution of the now defined “exposome” to extrinsic skin aging and skin conditions is well accepted. Pollution is one of the main players included in the skin exposome [224]. It has been recently shown [285] that exposure to more pollutants can have an additive effect. This could be a consequence of the different mechanisms of action of each stressor based on its chemical/physical properties.

It is generally understood that the toxic effects of O_3,_ although it is not a radical species, per se, are mediated through free radical reactions either directly by the oxidation of biomolecules to give classical radical species (hydroxyl radical) or by driving the radical-dependent production of cytotoxic nonradical species (aldehydes) [286].

O_3_ cannot penetrate the SC, so it first interacts with the lipids present in the outermost layer of the skin, leading to the generation of a number of bioreactive species [287]. It can be suggested that reaction with the well-organized interstitial lipids and protein constituents of the outermost stratum corneum barrier, and the diffusion of bioreactive products from this tissue into the viable layers of the epidermis, may represent a contribution to the development/exacerbation of skin disorders associated with O_3_ exposure. Indeed, once these “mediators” can reach live cells (keratinocytes, fibroblasts, etc.), they can induce a cellular defensive and inflammatory response that leads to an inflammatory/oxidative vicious cycle, called OxInflammation [288]. Unless quenched by endogenous or exogenous mechanisms, this will damage the skin and compromise its barrier functions, contributing to extrinsic skin aging.

Different hypotheses have been proposed concerning the initiation of the PM’s detrimental effects on cutaneous tissues. This could be due to an indirect effect by an outside-inside signaling cascade. PM, especially smaller particles, may carry metal ions and/or organic compounds such as polycyclic aromatic hydrocarbons (PAHs), which are highly lipophilic and can penetrate the skin surface [287]. This is in agreement with our observations. Moreover, PAHs are potent ligands for the AhR receptor, expressed by both keratinocytes and melanocytes, which upregulates proinflammatory mediators and increases ROS production [289].

In a previous study, we provided evidence that PM develops cutaneous damage not only directly once particles reach deeper layers in the epidermis but also indirectly by triggering a signaling pathway [290]. Oxidative stress and an inflammatory response seem to be important steps in the PM toxic mechanisms.

The solar spectrum reaching the surface of the earth is divided into three main segments based on wavelength: UVC (100−290 nm), UVB (290−320 nm), and UVA (320−400 nm). Both UVA and UVB have acute and chronic effects on human skin [287]. It has been established that approximately 50% of UV-induced direct cellular injury accounts for the remainder of the damage [291,292,293].

Therefore, although UV, PM, and ozone have different mechanisms of action, they all have the common denominator of damage that can be summarized as oxidative stress.

This effect is not limited to a biochemical effect, but it has been shown that air pollutants modulate epigenetic states, ranging from DNA methylation to miRNAs expression [294].

The aim of this study was to evaluate the different miRNA cutaneous responses to the main pollutants to which our skin is exposed daily.

It was not surprising that there was a clear difference among the pollutants in terms of the modulated miRNAs and the pathways associated with the epigenetic variation.

We found that the main pathways affected by the analyzed pollutants were: apoptosis, cell cycle, inflammation, DNA repair, and cancer. Of note, apoptosis was not associated with O_3_ exposure, and this could be a consequence of the O_3_ mechanism of action, which leads to the generation of proinflammatory mediators (H_2_O_2_ and aldehydes) less aggressively compared to UV and PM [89,100]

The ability of UV to induce DNA damage and subsequent apoptosis has been well demonstrated in the past, and our data confirm these results under the epigenetic mechanism as well. In addition, our data confirmed the involvement of O_3_ exposure in cutaneous inflammation, as previously demonstrated by Xu et al., where ER visit for skin inflammatory conditions perfectly correlated with increased ozone levels in urban cities [17,20].

To confirm that the O_3_ epigenetic effect is tissue-specific (due to the different mediators generated by the interaction with different tissues), the work by [295] showed that the expression analysis of sputum samples revealed that O_3_ exposure significantly increased the expression levels of several miRNAs, namely miR-132, miR-143, miR-145, miR-199b-5p, miR-222, miR-223, miR-25, miR-424, and miR-582-5p that while not detected in our analysis were still involved in inflammation. A quite recent work suggests that the main effect of PM on the skin is due to the absorption of PAHs, which can lead to skin barrier perturbation and damage [296]. PAH exposure has been already associated with epigenetic variation related mainly to DNA methylation.

In the present study, we have shown for the first time that PAH (present in diesel particles) can affect cutaneous epigenetics related to miRNA expression, highlighting the possible detrimental effect that those compounds can have on the skin [61]. The time-related kinetic differences in miRNA expression at 0 and 24 h reflect the different nature of the tested pollutants.

Ozone is a volatile gas; accordingly, its interaction with the skin induces effects on a short–term basis because this gas is neither metabolized nor entrapped into skin layers. UV radiation induces short-term alteration triggering reactive mechanisms (DNA repair, etc.), requiring at least 8 h to be activated by the modulation of the microRNA machinery. MicroRNAs are highly sensitive to environmental stressors, as is well demonstrated in the lung for cigarette smoke [297] and airborne pollutants [298]. However, this issue has not yet been explored in the skin. The presented results herein provide experimental evidence that human skin undergoes dramatic changes in its physiological microRNA profile when exposed to environmental pollutants, either physical or chemical.

Diesel fumes are well known to induce genotoxic damage and DNA adduct formation [299], as well as microRNA alteration in the lung [300]. Diesel extracts can induce cancer in mouse skin [301]. The carcinogenic effect of diesel fumes is mainly due to the presence in this mixture of potent chemical genotoxic carcinogens such as nitropyrenes [302]. Indeed, epigenome regulation performed by the microRNA machinery can silence the expression of mutated oncogenes, thus defending our organism from the progression of the carcinogenic process. Only when genomic damage accumulates in the presence of irreversible alteration of the microRNA machinery does cancer occur [303].

Accordingly, the demonstration that microRNA are dramatically altered in human skin shows that diesel fumes are a complete skin carcinogen inducing both genomic and epigenomic alterations. The carcinogenicity of diesel fumes is exerted by the phase I and II metabolic reaction of its chemical components with particular reference to polycyclic aromatic hydrocarbons. This situation, together with the limited metabolic potential of the skin as compared to other tissues, explains why microRNA alterations reach the maximum level only after 40 h of exposure. The carcinogenicity of diesel fumes is confirmed by the finding that the majority (40%) of altered microRNA is involved in carcinogenesis processes, while only a minority is involved in defensive processes such as apoptosis (22%) and DNA repair (4%) [73,76].

Ozone displays an immediate effect on skin microRNA that does not increase after 24 h from the exposure. This finding indicates that adaptive mechanisms are triggered by ozone. Skin is well equipped with antioxidant defenses requiring some time to be activated. This situation is confirmed by the finding that the majority (31%) of altered microRNA is involved in defensive mechanisms allowing removal of damaged cells (apoptosis) and DNA repair (8%, i.e., two-fold more than diesel fumes), while only a minority in carcinogenesis (20%, i.e., two-fold less than diesel fumes).

UV radiation is carcinogenic, as demonstrated by the finding that this exposure induces the most dramatic alterations of microRNA involved in carcinogenesis (45%) compared to diesel fume and ozone.

Furthermore, UV displays a variety of other adverse biological effects, including cell loss by apoptosis, cell proliferation to replace lost cells, and inflammation. In this regard, microRNA alteration overlaps the main function of genes whose expression undergoes upregulation, as demonstrated in mouse skin [299].

Conversely, only a minority (7%) of microRNA involved in DNA repair are activated in human skin after 24 and 48 h. This situation is different from those reported in vivo in mouse skin after long-term exposure that activated a variety of genes involved in base excision (XP) and nucleotide excision (OGG1) DNA repair [304]. This finding indicates that only long-term exposure to UV is effective in activating defensive DNA repair, while exposures to high doses for a short time results in a lack of DNA repair activation. Because of this situation, sunburns, indicating the occurrence of UV exposure in the absence of effective activation of DNA and protein repair, represent a major risk factor for cancer development.

A limitation of the presented study is that skin biopsies were collected from three subjects only. Future studies using a wider number of subjects are necessary to explore the interindividual variation occurring in the miRNA of human skin when exposed to environmental pollutants.

## 5. Conclusions

In conclusion, our results provide experimental evidence in human skin that microRNA machinery is altered by exposure to environmental pollutants. This situation occurs regarding either chemical pollutants, such as diesel fumes and ozone, or exciting radiation, such as UV. MicroRNA alteration, on a short-term basis, represents an adaptive event triggering defensive mechanisms such as DNA repair and apoptosis, attenuating the consequences of molecular damage induced in the skin by environmental stressors. Whenever microRNA alterations persist for a long time because of continuous exposure to environmental stressors, microRNA defensive function is neutralized, thus leaving the pathology or even carcinogenesis process to develop. Our findings demonstrate that skin has potent microRNA machinery used to face exposure to environmental pollutants. The alteration induced in skin microRNA undergoes a signature specific to the environmental pollutant involved.

## Figures and Tables

**Figure 1 jpm-12-00176-f001:**
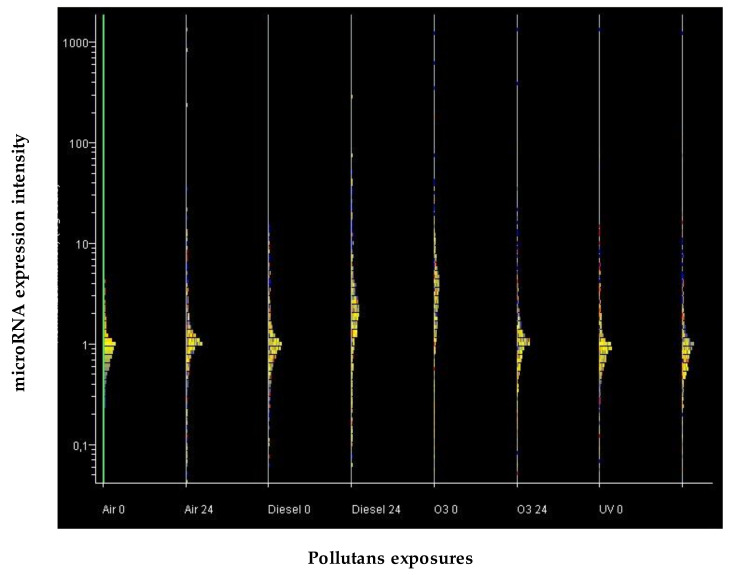
Line plot analysis of the overall expression of the 2549 human miRNAs analyzed under each experimental condition tested. The expression of 2549 human miRNAs was evaluated at 0 and 24 h in skin either unexposed (Air) or exposed to diesel, ozone, and UV. miRNAs are distributed in horizontal lines according to their level of expression, the majority being located at intermediate levels of expression (central part of the distribution), and the minority being located at high and low levels of expression (lower and upper part of the distribution). The distribution profile is progressively modified according to the treatment used.

**Figure 2 jpm-12-00176-f002:**
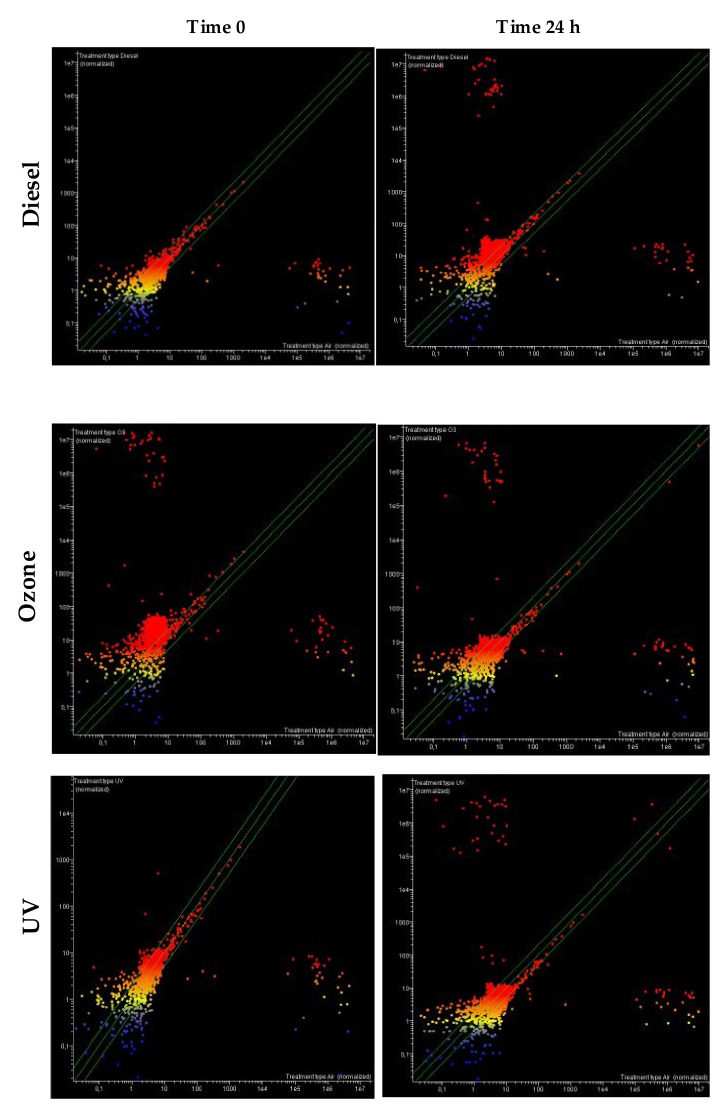
Scatter plot analysis of changes in miRNA-expression in human skin upon exposure to diesel, ozone, and UV at different times (0 and 24 h).

**Figure 3 jpm-12-00176-f003:**
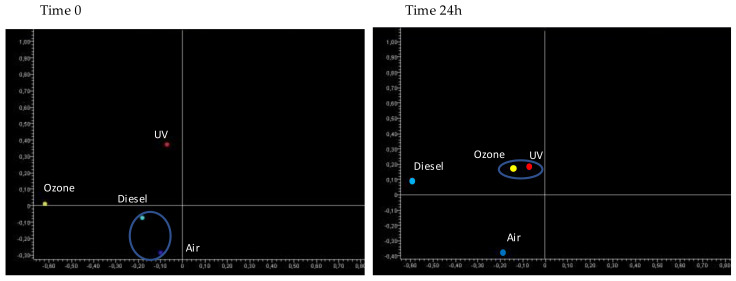
Bidimensional principal component analysis (PCA) of miRNA profiles of skin samples either untreated (air) or treated with diesel, ozone, and UV at 0 (**left panel**) and 24 h (**right panel**). PCA1 (X axis), PCA2 (Y axis).

**Figure 4 jpm-12-00176-f004:**
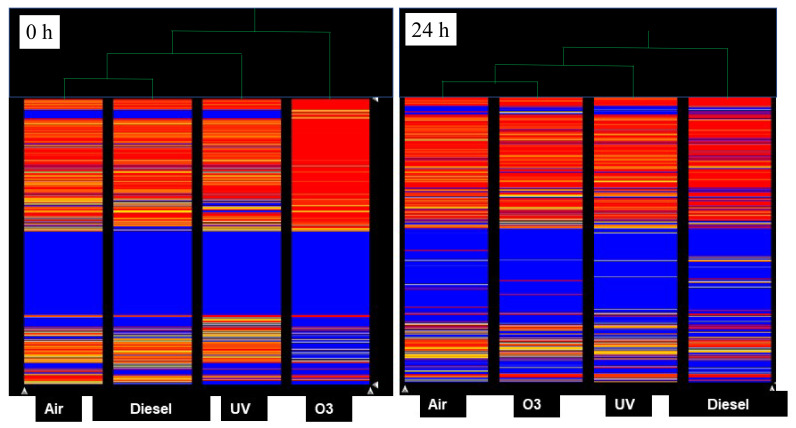
Hierarchical Cluster Analysis (HCA) reporting the expression of the 2549 miRNAs (colored horizontal bars) tested at time 0 (**left panel**) and 24 h (**right panel**) in samples either untreated (air) or treated with pollutants (ozone, diesel, and UV). Columns report miRNA expression profiles for each experimental condition. Similar expression profiles are linked in the upper hierarchical tree (green), thus being located nearby, while different expression profiles are located far away in the hierarchical tree.

**Table 1 jpm-12-00176-t001:** Diesel.

MicroRNA	Fold Change	Function	Reference
hsa-miR-495-5p	>10	Promotes Th2 differentiation in allergic rhinitis, as a tumor suppressor	[23]
hsa-miR-628-5p	>10	Inhibits osteoblast differentiation via RUNX2	[24]
hsa-miR-361-3p	>10	Suppresses proliferation, invasion inhibited cells invasive and proliferative abilities, and cell lines invasion and proliferation	[25,26]
hsa-miR-875-5p	>10	Promotes cellular apoptosis and proliferation	[27]
hsa-miR-509-3p	>10	Tumor suppressor	[28,29]
hsa-miR-518b	>10	Suppresses cell proliferation, invasiveness, and migration in colorectal cancer	[30,31]
hsa-miR-516b-5p	>10	Cell proliferation, inducing G1 cell cycle arrest and apoptosis	[32]
hsa-miR-381-5p	>10	Induces apoptosis	[33]
hsa-miR-661	>10	Promotes proliferation, migration, and metastasis of NSCLC	[34]
hsa-miR-216a-3p	>10	Antitumor functions	[35]
hsa-miR-548c-3p	>10	Inflammatory responses and potential estrogen receptor sensitivity	[36]
hsa-miR-106a-3p	>10	Cell proliferation and autophagy	[37]
hsa-miR-616-5p	>10	Promotes angiogenesis and modulates cell proliferation	[38]
hsa-miR-671-3p	>10	Suppresses proliferation and invasion of breast cancer cells and regulates metabolic processes	[39,40]
hsa-miR-544a	>10	Regulates migration and invasion in colorectal cancer cells	[41]
hsa-miR-614	>10	Inflammatory process	[42]
hsa-miR-525-3p	<10	Modifies the expression of proinflammatory cytokines; apoptosis	[43]
hsa-miR-378c	<10	Regulates the angiogenic capacity of CD34(+) progenitor cells	[44]
hsa-miR-924	<10	Suppresses the proliferation, migration, and invasion of NSCLC cells	[45]
hsa-miR-522-3p	<10	Cell proliferation of human glioblastoma cells; modulates the expression of proinflammatory cytokines	[46]
hsa-miR-431-5p	<10	Promotes differentiation and regeneration of cells	[47,48]
hsa-miR-770-5p	<10	Suppresses cell apoptosis and the release of proinflammatory factors	[49]
hsa-miR-183-5p	<10	Tumor suppressor inflammation and alters miRNA expression in the airway epithelium	[50]
hsa-miR-598-5p	<10	Promotes cell proliferation and cell cycle progression in human colorectal carcinoma; elevates apoptosis	[51]
hsa-miR-486-5p	<10	Regulation of heart contraction, muscle contraction, and ion channel activity	[52]
hsa-miR-125a-5p	<10	Regulates stress response, apoptosis, proliferation, angiogenesis, and expression of genes, associated with human lung cancer	[53]
hsa-miR-34b-5p	<10	Regulates stress response, apoptosis, proliferation, angiogenesis, and expression of genes, and is upregulated during cardiac hypertrophy	[54]
hsa-miR-301a-3p	<10	Promotes autophagy and inhibits apoptosis	[55,56]
hsa-miR-23c	<10	Inhibits cell proliferation and induces apoptosis of hepatocellular carcinoma cells; cell growth arrest and apoptosis.	[57]
hsa-miR-383-5p	<10	Oxidative stress and inflammation-related factors	[58]
hsa-miR-574-5p	<10	Promotes the differentiation of human cardiac fibroblasts	[59]
hsa-miR-151a-5p	<10	Regulation of cellular respiration and ATP production through targeting Cytb	[60]
hsa-miR-514a-3p	<10	Attenuates proliferation and increases chemoresistance	[61]
hsa-miR-136-3p	<10	Promotes apoptosis in gastric cancer cells	[62]
hsa-miR-1-3p	<10	Inflammation; regulator of heart adaption after ischemia or ischemic stress	[63]
hsa-miR-18a-3p	<10	Downregulated in aging cells; induces the apoptosis of colon cancer cells	[64]
hsa-miR-502-5p	<10	Enhances early apoptosis and inhibits proliferation of breast cancer cells	[65]
hsa-miR-451a	<10	Apoptosis; inhibits autophagy	[66]
hsa-miR-19b-1-5p	<10	Linked to oxidative stress, inflammation, and atherosclerosis	[67]
hsa-miR-873-5p	<0.1	Tumor suppressor in thyroid cancer by inhibiting the proliferation, migration, and invasion of the cancer cells	[68]
hsa-miR-212-5p	<0.1	Promotes cancer cell apoptosis and suppresses cancer cell proliferation and invasion	[69]
hsa-miR-552-5p	<0.1	Tumorigenesis; progression	[70]
hsa-let-7a-3p	<0.1	Linked to oxidative stress, inflammation, and atherosclerosis	[71]
hsa-miR-495-3p	<0.1	Regulates proliferation, apoptosis, migration, and invasion in metastatic prostate cancer cells.	[68]
hsa-miR-767-3p	<0.1	Promoted cell proliferation in human melanoma cell lines	[72]
hsa-miR-27b-5p	<0.1	Involved in beige and brown adipogenesis after cold exposure	[73]
hsa-miR-26a-2-3p	<0.1	Regulated stress response, apoptosis, proliferation, angiogenesis, and expression of genes	[74]
hsa-miR-629-5p	<0.1	Action on tumor growth and metastasis in hepatocellular carcinoma	[75]
hsa-miR-543	<0.1	Cell oxidative phosphorylation	[76]
hsa-miR-515-5p	<0.1	Upregulated in placentas from women with preeclampsia	[77]
hsa-miR-708-5p	<0.1	Induces apoptosis and suppresses tumorigenicity in renal cancer cells	[78]
hsa-miR-378j	<0.1	Tumor suppressor	[79]
hsa-miR-548az-3p	<0.1	Alters inflammation	[80]
hsa-miR-876-5p	<0.1	Regulates regulation proliferation, migration, invasion, and glutaminolysis in gastric cancer cells	[37]
hsa-miR-656-3p	<0.1	Suppresses glioma cell proliferation, neurosphere formation, migration, and invasion	[81]
hsa-miR-944	<0.1	Increases p53 expression in cancer cells	[82]
hsa-miR-518e-3p	<0.1	Tumor suppressor	[83]
hsa-miR-373-3p	<0.1	Promotes the invasion and migration of breast cancer; regulates inflammatory cytokine-mediated chondrocyte proliferation	[32]

**Table 2 jpm-12-00176-t002:** Ozone exposure.

MicroRNA	Fold Change	Function	Reference
hsa-miR-628-5p	>10	Inhibits osteoblast differentiation	[84]
hsa-miR-15a-3p	>10	Proliferation; inflammation; apoptosis	[25,26]
hsa-miR-548am-3p	>10	Induces proliferation and migration	[85]
hsa-miR-550b-2-5p	>10	Cancer promotion	[86]
hsa-miR-495-5p	>10	Tumor suppressor; proliferation and differentiation of osteoblasts in mice; inhibits the growth of fibroblasts in hypertrophic scar	[87]
hsa-miR-345-3p	>10	Apoptosis and inflammation	[24,88]
hsa-miR-548q	>10	Induces proliferation and migration	[89]
hsa-miR-887-3p	>10	Pathways in cancer	[86]
hsa-miR-877-5p	>10	Pathways in cancer	[90]
hsa-miR-513c-5p	>10	Pathways in cancer	[91]
hsa-miR-422a	>10	Pathways in cancer	[92]
hsa-miR-194-5p	>10	Pathways in cancer	[93]
hsa-miR-378b	>10	Inflammation and cell cycle	[94]
hsa-miR-610	>10	Pathways in cancer	[95]
hsa-miR-519e-5p	>10	Atherosclerosis and pathways in cancer	[96]
hsa-miR-627-5p	>10	Cell proliferation and cancer promotion	[97,98]
hsa-miR-548au-5p	>10	Pathways in cancer	[99]
hsa-miR-770-5p	>10	Apoptosis and inflammation	[100]
hsa-miR-196b-3p	>10	Cell proliferation	[101]
hsa-miR-330-3p	>10	Apoptosis and cell proliferation	[102]
hsa-miR-617	>10	Pathways in cancer	[103]
hsa-miR-375	>10	Cell proliferation and pathways in cancer	[104]
hsa-miR-936	>10	Cell proliferation, pathways in cancer, and apoptosis	[105]
hsa-miR-657	>10	Inflammation	[106]
hsa-miR-542-5p	>10	Mitochondrial dysfunction and inflammation	[107]
hsa-miR-136-3p	>10	Vascularization and pathways in cancer	[108]
hsa-miR-409-5p	>10	Cardiovascular process, proliferation, migration, and cell cycle.	[109,110]
hsa-miR-154-3p	>10	Pathways in cancer	[111,112]
hsa-miR-378c	>10	Proliferation and inhibited apoptosis	[113]
hsa-miR-93-3p	>10	Inflammation and apoptosis	[114]
hsa-miR-556-3p	>10	Cell proliferation and apoptosis	[115]
hsa-miR-518c-5p	>10	Tumor suppressor	[116]
hsa-miR-23b-5p	>10	Cell proliferation and cancer	[32]
hsa-miR-504-5p	>10	Cell proliferation and differentiation	[117]
hsa-miR-509-3p	>10	Cardiovascular process	[118]
hsa-miR-514a-3p	>10	Tumor suppressor	[119]
hsa-miR-431-5p	>10	Cell proliferation and apoptosis	[120]
hsa-miR-506-3p	>10	Cell proliferation and cancer	[121]
hsa-miR-645	>10	Cell proliferation and cancer	[122]
hsa-miR-129-5p	>10	Inhibits the proliferation and metastasis of gastric cancer cells	[123]
hsa-miR-516b-5p	>10	Migration, cell proliferation, and cancer process	[124]
hsa-miR-512-3p	>10	Apoptosis and cell cycle	[125]
hsa-miR-101-5p	>10	Apoptosis and promotes cell proliferation	[126]
hsa-miR-561-5p	>10	Cell proliferation, G(1)/S transition, and suppresses apoptosis	[127]
hsa-miR-194-5p	>10	Cell proliferation and cancer	[128]
hsa-miR-329-3p	>10	Proliferation, invasion, and suppresses cell apoptosis	[129]
hsa-let-7i-3p	>10	Coronary disease and cancer	[130]
hsa-miR-129-2-3p	>10	Proliferation, invasion, and apoptosis	[131]
hsa-miR-548a-5p	>10	Proliferation and inhibits apoptosis	[132]
hsa-miR-887-5p	>10	Pathways in cancer	[133]
hsa-miR-99a-3p	>10	Cell proliferation and pathways in cancer	[134]
hsa-miR-487a-3p	>10	Cell proliferation and cancer	[135]
hsa-miR-378g	>10	Cancer promotion	[136]
hsa-miR-548at-5p	>10	Neurodegenerative disease	[137]
hsa-miR-374c-5p	>10	Proliferation, apoptosis, and autophagy	[138]
hsa-miR-106a-3p	>10	Proliferation and apoptosis	[139]
hsa-miR-92a-2-5p	>10	Apoptosis and cell proliferation	[140]
hsa-miR-616-5p	>10	Invasion, cell migration, and cancer	[141]
hsa-miR-509-5p	>10	Tumor suppressor	[142]
hsa-miR-598-3p	>10	Cancer process	[21]
hsa-miR-873-5p	>10	Cell migration and cancer	[143]
hsa-miR-525-3p	<10	Cancer cell migration	[21]
hsa-miR-500a-5p	<10	Cell apoptosis and proliferation	[144]
hsa-miR-659-3p	<10	Cell proliferation and cancer; apoptosis	[145]
hsa-miR-526b-5p	<10	Cell proliferation and cancer	[146]
hsa-miR-764	<10	Cardiac diseases and cancer	[147]
hsa-miR-934	<10	Cancer progression and inflammation	[148]
hsa-miR-516a-5p	<10	Cell proliferation and cancer	[149]
hsa-miR-520f-3p	<10	DNA repair	[150]
hsa-miR-369-5p	<10	Aerobic glycolysis and pathways in cancer	[151]
hsa-miR-613	<10	Invasion and cell proliferation	[152]
hsa-miR-411-3p	<10	Proliferation and cancer	[153]
hsa-miR-432-3p	<10	Inflammation	[154]
hsa-let-7c-3p	<10	Apoptosis	[155]
hsa-miR-671-5p	<10	Proliferation and cell cycle	[156]
hsa-miR-181d-5p	<10	Proliferation and angiogenesis	[157]
hsa-miR-192-5p	<10	Hypertension and cancer	[158]
hsa-let-7g-3p	<10	Linked to oxidative stress, inflammation, and atherosclerosis	[159]
hsa-miR-1-3p	<10	Decreases tumor volume in a xenograft model	[68]
hsa-miR-515-3p	<10	Cell proliferation, migration, invasion, and induced apoptosis	[160]
hsa-miR-320d	<10	Apoptosis and cancer	[161]
hsa-miR-548aa	<10	Can alter the inflammatory responses	[162]
hsa-miR-502-5p	<10	Cell proliferation and invasion	[37]
hsa-miR-758	<10	Proliferation and invasion	[163]
hsa-miR-7-1-3p	<10	Autophagy and cancer process	[164]
hsa-miR-324-3p	<10	Cell proliferation and cancer	[165]
hsa-miR-520g-3p	<10	DNA repair	[166]
hsa-miR-576-5p	<10	Cell invasion and cancer	[151]
hsa-miR-520a-3p	<10	Inhibits tumor progression, indicating its potential role as a tumor suppressor.	[167]
hsa-miR-449b-3p	<10	Proliferation	[168]
hsa-miR-211-5p	<10	Pathways in cancer	[169]
hsa-miR-376a-3p	<10	Coronary artery disease	[170]
hsa-miR-939-3p	<10	Pathways in cancer	[171]
hsa-miR-214-3p	<10	Inhibition of migration and proliferation	[172]
hsa-miR-609	<10	Cardiovascular process	[173]
hsa-miR-29a-5p	<10	Cardiac myocytes and overall cardiac dysfunction	[174]
hsa-miR-449c-3p	<10	Inhibits NSCLC cell progression	[175]
hsa-miR-185-3p	<10	Proliferation and invasion of cell	[176]
hsa-miR-766-3p	<10	Suppresses apoptosis and facilitates autophagy	[177]
hsa-miR-486-5p	<10	Regulation of heart contraction, muscle contraction, and ion channel activity	[53]
hsa-miR-144	<10	Tumor inhibitors or tumor suppressors, proliferation, and apoptosis	[53]
hsa-miR-664a-5p	<10	Induces cell differentiation	[178]
hsa-miR-32-3p	<10	Atherosclerosis	[179]
hsa-miR-224-3p	<10	Cell proliferation and promotes apoptosis	[180]
hsa-miR-130a-5p	<10	Myocardial infarction	[181]
hsa-miR-378i	<10	Metabolic pathways, mitochondrial energy homeostasis, and related biological processes	[182]
hsa-miR-642b-5p	<10	Inflammation	[183]
hsa-miR-668-3p	<10	Progression of different types of cancer	[184]
hsa-miR-18b-5p	<10	Progression of different types of cancer; cardiac function	[185]
hsa-miR-483-3p	<10	Apoptosis	[186]
hsa-miR-485-3p	<10	Cell proliferation; pathways in cancer	[187]
hsa-miR-200c-5p	<10	Oxidative stress and cell apoptosis	[188]
hsa-miR-126-5p	<10	Linked to oxidative stress, inflammation, and atherosclerosis	[189]
hsa-miR-26b-3p	<10	Cell proliferation and invasion	[190]
hsa-miR-378d	<10	Proliferation and migration of cancer	[191]
hsa-miR-526b-3p	<10	Regulates the proliferation, invasion, and migration of cancer cells	[192]
hsa-miR-575	<10	Oncogene	[193]
hsa-miR-564	<10	Cell proliferation and invasion	[194]
hsa-miR-513a-5p	<10	Induced apoptosis	[195]
hsa-miR-548i	<10	Downregulates the inflammatory cytokines	[196]
hsa-miR-188-5p	<10	Cell proliferation and cancer promotion	[197]
hsa-miR-563	<10	Cell proliferation and cancer promotion	[198]
hsa-miR-139-3p	<10	Proliferation and invasion	[199]
hsa-miR-34a-5p	<0.1	Inflammation	[200]
hsa-miR-34b-5p	<0.1	P53 effector, cell proliferation, and apoptosis	[201]
hsa-miR-371b-5p	<0.1	Cell proliferation and apoptosis	[202]
hsa-let-7f-2-3p	<0.1	Cell proliferation and apoptosis	[203,204]
hsa-miR-557	<0.1	Tumor suppressor	[205]
hsa-miR-574-5p	<0.1	Cell cycle and cancer process	[206]
hsa-miR-216a-3p	<0.1	Cell proliferation and apoptosis	[207]
hsa-miR-466	<0.1	Tumor suppressor	[208]
hsa-miR-222-3p	<0.1	Cell viability, migration, and invasion	[209]
hsa-miR-586	<0.1	Cell proliferation, invasion, metastasis, and apoptosis	[210]
hsa-miR-939-5p	<0.1	Inflammation	[211]
hsa-miR-548b-3p	<0.1	Proliferation, apoptosis, and mitochondrial function	[212]
hsa-miR-517c-3p	<0.1	Responses to stress; alterations in circulating glucose levels	[213]
hsa-miR-630	<0.1	Oxidative damage and cell migration	[214]
hsa-miR-544a	<0.1	Pathways in cancer	[215]
hsa-miR-603	<0.1	Proliferation, migration, invasion, and metastasis	[216]
hsa-miR-552-5p	<0.1	Cell proliferation	[217]
hsa-miR-562	<0.1	Tumor suppressor	[218]
hsa-miR-548	<0.1	Cancer cell proliferation, migration, and invasion	[219]
hsa-miR-518a-	<0.1	Tumor suppressor	[220]
hsa-miR-433-5p	<0.1	Cardiovascular process	[32]
hsa-miR-138-5p	<0.1	Cardiac function and pathological damage	[221]
hsa-miR-548ad-5p	<0.1	Cancer cell proliferation, migration, and invasion	[222]
hsa-miR-450a-2-3p	<0.1	Cell proliferation and cancer	[220]
hsa-miR-548av-3p	<0.1	Cancer cell proliferation, migration, and invasion	[223]
hsa-miR-624-5p	<0.1	Cell proliferation and cancer	[220]
hsa-miR-553	<0.1	Cell proliferation and cancer	[224]
hsa-miR-876-5p	<0.1	Cell proliferation and cancer	[225]
hsa-miR-190b	<0.1	Autophagy and cell cycle	[226]
hsa-miR-26a-2-3p	<0.1	Cell cycle and cancer process	[227]
hsa-miR-515-5p	<0.1	Cardiac function and proliferation cells	[75]
hsa-miR-195-3p	<0.1	Cardiac function and proliferation cells	[228]
hsa-miR-365b-5p	<0.1	Inflammation and cell proliferation	[229]
Hsa-miR-885-3p	<0.1	Inflammation	[230]

**Table 3 jpm-12-00176-t003:** UV exposure.

MicroRNA	Fold Variation	Functions	Reference
hsa-miR-329-3p	>10	Inhibits cell proliferation in glioma cells	[231]
hsa-miR-520g-5p	>10	DNA repair	[232]
hsa-miR-216a-	>10	Regulates the proliferation, apoptosis, migration, and invasion of lung cancer cells	[151]
hsa-miR-548c-3p	>10	Cancer cell proliferation, migration, and invasion	[233]
hsa-miR-129-2-3p	>10	Inhibits the proliferation and metastasis of gastric cancer cells	[220]
hsa-miR-887-5p	<10	Pathways in cancer	[124]
hsa-miR-106a-3p	>10	Involved in tumorigenesis and highly expressed in gastric cancer	[234]
hsa-miR-616-5p	>10	Progression of bladder cancer by regulating cell proliferation, migration, and apoptosis	[235]
hsa-miR-509-5p	>10	Tumor suppressive effects	[236]
hsa-miR-648	>10	Post-transcriptional regulators of glioblastoma	[237]
hsa-miR-378h	>10	Metabolic pathways, mitochondrial energy homeostasis, and angiogenic network in tumors	[238]
hsa-miR-200c-5p	>10	Upregulated by oxidative stress and induces endothelial cell apoptosis	[183]
hsa-miR-525-3p	<10	Pathways in cancer	[239]
hsa-miR-488-3p	<10	Pathways in cancer	[44]
hsa-miR-101-5p	<10	Regulates cell proliferation	[240]
hsa-let-7g-3p	<10	Modulates inflammatory responses; pathways in cancer	[241]
hsa-miR-376a-2-5p	<10	Pathways in cancer	[242]
hsa-miR-640	<10	Pathways in cancer	[243]
hsa-miR-300	<10	Controls stem cell function and inhibits differentiation	[244]
hsa-miR-509-3p	<10	Pathways in cancer	[245]
hsa-miR-548au-5p	<10	Pathways in cancer	[246]
hsa-miR-337-3p	<10	Pathways in cancer	[100]
hsa-miR-411-3p	<10	Pathways in cancer	[247]
hsa-miR-494-3p	<10	Mitochondrial biogenesis and pathways in cancer	[248]
hsa-let-7e-3p	<10	Pathways in cancer	[249]
hsa-miR-144-3p	<10	Regulates adipogenesis and pathways in cancer	[250]
hsa-miR-196b-3p	<10	Cell proliferation	[251]
hsa-miR-10b-5p	<10	Pathways in cancer	[102]
hsa-miR-33a-5p	<10	Associated with carcinogenesis	[252]
hsa-miR-136-3p	<10	Cardiac function and pathological damage in myocardial tissue, cardiomyocyte apoptosis, oxidative stress, and inflammatory response	[253]
hsa-miR-143-3p	<10	Pathways in cancer	[254]
hsa-miR-762	<10	Modulates thyroxine-induced cardiomyocyte and pathways in cancer	[255]
hsa-miR-582-5p	<10	Cell proliferation	[256]
hsa-miR-645	<10	Cell proliferation and apoptosis	[21]
hsa-miR-411-5p	<10	Cell proliferation and pathways in cancer	[257]
hsa-miR-7-1-3p	<10	Inhibits autophagy and induces apoptosis in glioblastoma	[258]
hsa-miR-18a-3p	<10	Progression of different types of cancer; cardiac function.	[259]
hsa-miR-99a-3p	<10	Cell proliferation and pathways in cancer	[186]
hsa-miR-491-5p	<10	Cell proliferation and pathways in cancer	[135]
hsa-miR-19b-1-5p	<10	Pathways in cancer	[260,261]
hsa-miR-614	<10	Cell proliferation and pathways in cancer	[262]
hsa-miR-493-5p	<0.1	Cell proliferation	[263]
hsa-miR-15b-3p	<0.1	Proliferation, inflammation, and apoptosis	[264]
hsa-miR-510-5p	<0.1	Cell proliferation	[265]
hsa-miR-485-5p	<0.1	Cell proliferation and pathways in cancer	[266]
hsa-miR-581	<0.1	Induces proliferation and migration	[188]
hsa-miR-340-3p	<0.1	Cell proliferation and pathways in cancer	[267]
hsa-miR-708-5p	<0.1	Cell proliferation and pathways in cancer	[268]
hsa-miR-548j-3p	<0.1	Induces proliferation and migration	[269]
hsa-miR-618	<0.1	Cell proliferation and pathways in cancer	[270]
hsa-miR-885-3p	<0.1	Inflammatory response and pathways in cancer	[271]
hsa-miR-297	<0.1	Oncogene, inflammatory response, and apoptosis	[231]
hsa-miR-518a-5p	<0.1	Tumor suppressor	[272,273]
hsa-let-7f-2-3p	<0.1	Proliferation and apoptosis	[32]
hsa-miR-519b-3p	<0.1	Radiosensitivity of radio-resistant cells and pathways in cancer	[205]
hsa-miR-27b-5p	<0.1	Prevents atherosclerosis by inhibiting inflammatory responses	[274]
hsa-miR-625-5p	<0.1	Inflammation and inhibits cardiac hypertrophy	[275]
hsa-miR-548ah-5p	<0.1	Induces proliferation and migration	[276]
hsa-miR-892c-3p	<0.1	Pathways in cancer	[86]
hsa-miR-373-3p	<0.1	Inhibits autophagy	[277]
hsa-miR-656-3p	<0.1	Induces proliferation and migration	[278]
hsa-miR-20a-3p	<0.1	Proliferation and autophagy	[279]
hsa-miR-518a-3p	<0.1	Tumor suppressor	[280]
hsa-miR-649	<0.1	Pathways in cancer	[32]
hsa-miR-483-3p	<0.1	Pathways in cancer and cardiac response	[281]
hsa-miR-501-3p	<0.1	Cell proliferation, clonogenicity, migration, and invasion	[282]
hsa-miR-335-5p	<0.1	Cell proliferation, migration, and invasion.	[283]
hsa-miR-129-5p	<0.1	Proliferation, invasion, and apoptosis	[284]
hsa-miR-34c-5p	<0.1	Proliferation and apoptosis	[132]
hsa-miR-548ao-5p	<0.1	Promotes proliferation and inhibits apoptosis	[202]
hsa-miR-624-5p	<0.1	Cell proliferation and pathways in cancer	[133]

**Table 4 jpm-12-00176-t004:** Number of the miRNAs involved in the pathways deregulated by each pollutant tested. Only miRNAs demonstrated to be involved in the pathways are included.

	Diesel	Ozone	UV
Apoptosis	16	36	12
Cell cycle	7	24	11
Inflammation	11	21	7
DNA repair	1	4	2
Pathways in cancer	23	72	39

## Data Availability

The datasets used and/or analyzed during the current study are available in Gene Expression Omnibus (http://www.ncbi.nlm.nih.gov/geo); GEO number accession requested, 15 March 2022).

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
