# Peer review of "MicroRNA Alterations Induced in Human Skin by Diesel Fumes, Ozone, and UV Radiation"

_jpm, 2022, doi:10.3390/jpm12020176_

Round 1

Reviewer 1 Report

Although the work presented in this research article is interesting and the experimental part has been conducted properly, there are serious flaws in English edit, materials and methods section and in the presentation of the results (inconsistence between figures and the text, insufficient information about the tables, not information about up regulation or down regulation of the different miRNAs, ridiculous fold variations of 11exp-10) as well as repetition of paragraphs.

Author Response

Dear Editors,

We would like to thank you for considering the manuscript entitled: “MicroRNA Alterations Induced in Human Skin by Diesel Fume, Ozone, and UV Radiation” by Valacchi G. et al., and for sharing the Reviewers’ comments that certainly helped in improving the quality of the manuscript (Manuscript ID: jpm-1430831). We appreciated the Reviewers’ comments, and we revised the manuscript accordingly.

Please find enclosed to the submission of the revised version of the manuscript the point-by point reply to the Reviewers’ comments. For clarity’s sake, changes in the revised MS are marked in yellow.

We hope that the revised version of our MS will be now suitable for publication in the Journal of Personalized Medicine.  

Accordingly, we prepared a revised version of the manuscript acknowledging Referees’ and Editor’s comments as below specified:

Reviewer 1:

COMMENT 1 Although the work presented in this research article is interesting and the experimental part has been conducted properly, there are serious flaws in English edit, materials and methods section and in the presentation of the results (inconsistence between figures and the text, insufficient information about the tables, not information about up regulation or down regulation of the different miRNAs, ridiculous fold variations of 11exp-10) as well as repetition of paragraphs.

ANSWER 1. We appreciated the Reviewers’ comments, and we revised the paper accordingly reviewer suggestions. The English language has been revised by the authors who worked in the USA for many years Prof. Valacchi G and Pambianchi E. Materials and methods section has been revised as requested and the presentation of results has been corrected. The repetitions of the paragraph have been modified. The fold variations in the tables have been corrected. All the required corrections have been made. We thank the reviewers for their comments. 

Reviewer 2 Report

This is a very well written manuscript, and interesting to read. Authors did a comprehensive experiment and analysis of three main pollutants in miRNA changes in human skin. Authors are well summarized and discussed the results. Few comments are listed as below,

  1. Have the authors thought about adjusting the strength of each exposure to represent different regional area, for example which people live in high UV exposure area, or high air polluted area?
  2. Have the authors thought about if the skin pigments would infect the miRNA alteration?

Author Response

Reviewer 2:

COMMENT 1This is a very well written manuscript, and interesting to read. Authors did a comprehensive experiment and analysis of three main pollutants in miRNA changes in human skin. Authors are well summarized and discussed the results. Few comments are listed as below:

ANSWER 1. We thank the reviewer for the positive comments, and we have addressed his/her comments as follow:

COMMENT 2. Have the authors thought about adjusting the strength of each exposure to represent different regional area, for example which people live in high UV exposure area, or high air polluted area?

ANSWER 2. We thank you the reviewer for the nice suggestion and we agree with it. Indeed, the follow up of this study will aim to understand not only UV dosage effect based on the geographical area but also different kind of particulates to which humans are exposed, as we know that there are some differences between motor engines and pollution sources among continents and countries. Of course, it is not always simple to mimic in a laboratory the environment conditions, so it is a good practice to first start simple and then, based on the results, increase the variables of the experimental procedure.

COMMENT 3 Have the authors thought about if the skin pigments would infect the miRNA alteration?

ANSWER 3. We thank the reviewer for this good point since it’s well known that cutaneous miRNA can regulate melanocortin receptors. It is possible that we influence pigmentation, although we did not observe it, and in a general concept we can include this effect as part of pollution influence skin miRNA expression and overall skin responses to pollution.

Round 2

Reviewer 1 Report

In this article the authors investigate the effects of three pollutants (UV light, ozone and diesel particulate material) in the miRNA expression profiles of skin samples of three individuals.

Although the introduction and the discussion are too extensive, the work presented in this research article is interesting and the experimental part has been conducted properly. After a first revision, some previous comments have not been considered. Still some flaws in the presentation of the results persist (interpretation of figures instead of data, insufficient information about the tables, not information about upregulation or downregulation of the different miRNAs, ridiculous fold variations) and need to be corrected. The next comments need to be addressed in deep.

Comments:

  1. Edition of the text is confusing and introduced new English edit mistakes.
  2. The authors state raw data from microarrays are deposited in Gene Expression Omnibus (NCBI) but in the Data Availability Statement they claim they will be shared by authors “on reasonable request”. What is the final position of the authors with respect to that?
  3. Bioinformatic analyses related to microarrays do not include any statistical analysis. The authors use volcano plots to select miRNAs differentially expressed (?!) with a threshold fold change of 2 and a p-value lower than 0.05. This threshold does not reflect the lists of miRNAs presented in tables 1, 2 and 3. Could the authors explain it?
  4. In line 217 the authors refer to tumoral tissues. Could the authors explain that?
  5. In line 220 the authors refer to five different environmental exposures. This sentence is not clear.
  6. Figures 1, 2, 3 and 4 have poor resolution and a black background. This has to be addressed.
  7. Rephrase lines 234 to 236.
  8. In figure 2, the authors use colours depending on the signal intensity. Could the authors explain it? Does it refer to any specific condition?
  9. In lines 289, 295 and 296, why downregulation of miRNAs is associated to low intensity signal? Why upregulated miRNAs are associate to high intensity signal? Is there any correlation?
  10. In figure 4 the hierarchical trees are not visible. This needs to be fixed.
  11. In line 346 a Venn diagram analysis is performed. This is confusing. A Venn diagram is a representation of data. Data are analysed to plot a diagram not the opposite.
  12. Why are miRNAs dysregulated at time 0 considered for the specific miRNA signature of each pollutant (line 347) if diesel and UV produce a clearer effect after 24 hours (lines 254-258)?
  13. Tables 1, 2 and 3 show a fold change. Is it with respect to untreated samples at time 0, time 24h? Is it between time 0 and 24 hours of each pollutant?
  14. Tables 1, 2 and 3 do not show p-values. Could the authors explain it?
  15. Tables 1, 2 and 3 do not show if the change in expression of the miRNAs was positive or negative. Could the authors explain it?
  16. Tables 1, 2 and 3 show miRNAs with expression fold change lower than 2. Could the authors explain it?
  17. Sentence in line 410.
  18. In line 411 the authors state that light coming form the sun has wavelengths below 400nm. What are the authors based on?
  19. Lines 438, 441
  20. Sentence in lines 456-459 needs a reference.
  21. Sentences in lines 481 to 485 need references.
  22. Sentence in lines 509-512 needs a reference.
  23. Nothing is said about the fact that only three individuals participated in the study. A sentence or two about the limitations of the study are also needed.

Author Response

Dear Editors,

We would like to thank you for considering the manuscript entitled: “MicroRNA Alterations Induced in Human Skin by Diesel Fume, Ozone, and UV Radiation” by Valacchi G. et al., and for sharing the Reviewers’ comments that certainly helped in improving the quality of the manuscript (Manuscript ID: jpm-1430831). We appreciated the Reviewers’ comments, and we revised the manuscript accordingly.

Please find enclosed to the submission of the revised version of the manuscript the point-by point reply to the Reviewers’ comments. For clarity’s sake, changes in the revised MS are marked in yellow.

We hope that the revised version of our MS will be now suitable for publication in the Journal of Personalized Medicine.  

Accordingly, we prepared a revised version of the manuscript acknowledging Referees’ and Editor’s comments as below specified:

Reviewer 1:

COMMENT 1 Although the work presented in this research article is interesting and the experimental part has been conducted properly, there are serious flaws in English edit, materials and methods section and in the presentation of the results (inconsistence between figures and the text, insufficient information about the tables, not information about up regulation or down regulation of the different miRNAs, ridiculous fold variations of 11exp-10) as well as repetition of paragraphs.

ANSWER 1. We appreciated the Reviewers’ comments, and we revised the paper accordingly reviewer suggestions. The English language has been revised by the MPDI English Language Service.  Materials and methods section has been revised as requested and the presentation of results has been corrected. The repetitions of the paragraph have been modified. The fold variations in the tables have been corrected. All the required corrections have been made. We thank the reviewers for their comments.